# Diversification of African Rainforest Restricted Clades: Piptostigmateae and Annickieae (Annonaceae)



**Baptiste Brée [1], Andrew J. Helmstetter [1], Kévin Bethune [1], Jean-Paul Ghogue [2], Bonaventure Sonké [2] and Thomas L. P. Couvreur [1,\***

[1] IRD, DIADE, CIRAD, University of Montpellier, 34090 Montpellier, France; baptiste.bree@etu.umontpellier.fr (B.B.); andrew.j.helmstetter@gmail.com (A.J.H.); kevin.bethune@hotmail.fr (K.B.)

[2] Plant Systematic and Ecology Laboratory, Department of Biology, Higher Teachers' Training College, University of Yaoundé I, P.O. Box 047, Yaoundé, Cameroon; jpghogue@greenconnexion-cm.org (J.-P.G.); bonaventuresonke@ens.cm (B.S.)

\* Correspondence: Thomas.Couvreur@ird.fr

**Abstract:** African rainforests (ARFs) are species rich and occur in two main rainforest blocks: West/Central and East Africa. This diversity is suggested to be the result of recent diversification, high extinction rates and multiple vicariance events between west/central and East African forests. We reconstructed the diversification history of two subtribes (Annickieae and Piptostigmateae) from the ecologically dominant and diverse tropical rainforest plant family Annonaceae. Both tribes contain endemic taxa in the rainforests of West/Central and East Africa. Using a dated molecular phylogeny based on 32 nuclear markers, we estimated the timing of the origin of East African species. We then undertook several diversification analyses focusing on Piptostigmateae to infer variation in speciation and extinction rates, and test the impact of extinction events. Speciation in both tribes dated to the Pliocene and Pleistocene. In particular, *Piptostigma* (13 species) diversified mainly during the Pleistocene, representing one of the few examples of Pleistocene speciation in an African tree genus. Our results also provide evidence of an ARF fragmentation at the mid-Miocene linked to climatic changes across the region. Overall, our results suggest that continental-wide forest fragmentation during the Neogene (23.03–2.58 Myr), and potentially during the Pliocene, led to one or possibly two vicariance events within the ARF clade Piptostigmateae, in line with other studies. Among those tested, the best fitting diversification model was the one with an exponential speciation rate and no extinction. We did not detect any evidence of mass extinction events. This study gives weight to the idea that the ARF might not have been so negatively impacted by extinction during the Neogene, and that speciation mainly took place during the Pliocene and Pleistocene.

**Keywords:** biogeographic vicariance; extinction; phylogenomics; gene shopping; gene capture; molecular dating

## 1. Introduction

Tropical rainforests are the most biodiverse terrestrial ecosystems in the world, despite covering less than 10% of the land surface [1,2]. African rainforests (ARFs) are one of the world's most biodiverse regions [3–5] and the Congo basin contain the second largest continuous expanse of this biome after the Amazon basin. Understanding why and when the ARF flora diversified has been the subject of several studies [6], which highlighted two main geological periods that could have been associated with increased speciation rates in ARF clades: (1) Late Oligocene–Miocene, characterized by climatic fluctuations and geological events that caused the fragmentation and re-expansion of ARFs, leading to

increased allopatric speciation [7–10]; (2) the Pleistocene, with paleobotanical studies suggesting that rapid climatic changes linked to the Milankovitch cycles enhanced allopatric speciation by isolating ARFs into small patches (lowland forest refugia) [11–14].

In addition, even though ARFs are species rich, they are less diverse than other major tropical rainforested regions such as the Neotropics and South East Asia, a pattern of tropical biodiversity referred to as the "odd man out" [15]. Numerous hypotheses have been suggested to explain these differences (reviewed by [16]). One hypothesis is linked to differences in diversification rates between regions, with tropical Africa undergoing higher levels of extinction rates connected to increased aridification since the Miocene [2,16–18]. Inferring diversification variation within rainforest restricted plant clades spanning most of the Cenozoic can provide a way to unravel the importance of the Miocene and Pleistocene periods, and also help understand the possible impact of past extinction history on present ARF diversity.

In parallel to the above, dated molecular phylogenies have also provided insights into ARF cover history since the Oligocene. Today, ARFs are divided into two main disjunct regions separated by a 1000 km wide arid corridor running from north to south in East Africa [9,19]: the Guineo–Congolian region (GC, located in west and central Africa) and the East African region (EA). In the former, forests extend from Liberia to the Republic of Congo and in vast areas of the Congo Basin. In the latter, rainforests occur in small patches along the East African coast and the Eastern Arc mountains of Kenya and Tanzania, and reaching northern Mozambique (Figure 1). Many floristic affinities have been suggested between these two biogeographic regions: many genera are widespread and common to both regions but at the specific level, most species are endemic either to GC or EA. In addition, each region is home to several endemic genera [9,20–26]. These affinities suggest that rainforests were once widespread across the continent, leading to a pan-African rainforest. Numerous studies suggested that ARFs could have stretched across the width of the continent during Paleocene and Eocene [2,27–29] (but see [30]).

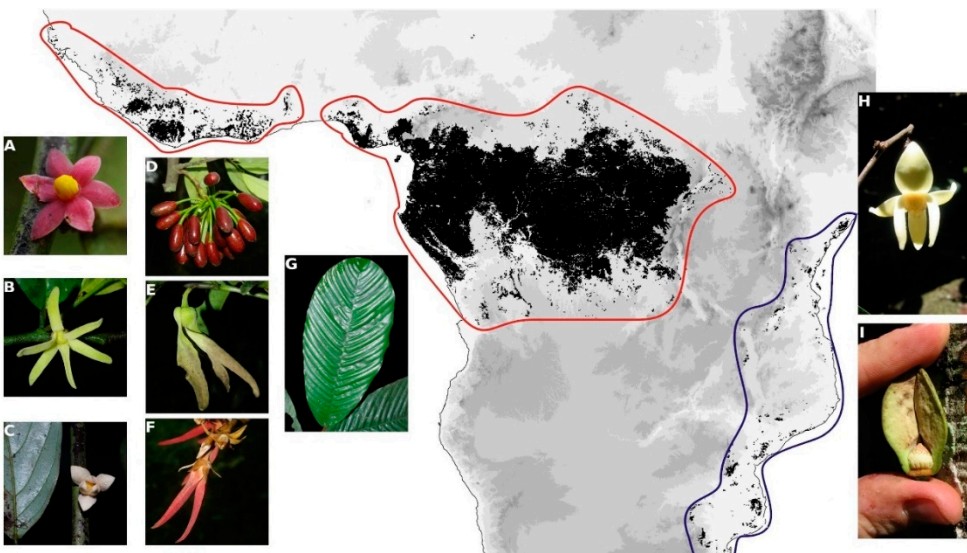

**Figure 1.** Present day distribution of tropical African rainforests separated between: (in red) the Guineo–Congolian region occurring in West to central Africa, and (in blue) the East African coastal and Eastern Arc rainforests. Map adapted from Couvreur et al. (2008) [9]. Photos represent different species from the Piptostigmateae tribe. A to G: endemic species to the Guineo–Congolian rainforests; (**H**, **I**) endemics to the east African rainforests. (**A**) *Sirdavidia solannona*, flower; (**B**) *Greenwayodendron suaveolens*, flower; (**C**) *Polyceratocarpus parviflorus*, flower; (**D**) *Annickia affinis*, fruit; (**E**) *Annickia affinis*, flower; (**F**) *Piptostigma multinervium*, flower; (**G**) *Piptostigma pilosum*, leaf, note numerous secondary veins; (**H**) *Mwasumbia alba*, flower; (**I**) *Annickia kummeriae*, flower. Photos: Thomas L. P. Couvreur.

Two hypotheses have been proposed as to how these disjunct patterns emerged: (1) the ARF fragmented only once during the Oligocene or Early Miocene due to global aridification following climatic changes of the Eocene/Oligocene Transition (EOT) around 33 Million years ago (Myr) [17,23,31], (2) ARFs have fragmented repeatedly since the Oligocene with phases of continental expansion and retraction linked to major climatic events [2,22,32,33]. Using a dated molecular phylogeny, Couvreur et al. (2008) [9] showed support for the second hypothesis, as they linked multiple vicariance speciation events with known periods of aridification and geological events in the rainforest restricted clade of African Annonaceae (tribe Monodoreae), revealing three possible fragmentation events during the Miocene and Oligocene epochs (at ca. 33, ca. 16 and ca. 8 Myr). Most other molecular dating studies published to date also support the hypothesis of multiple fragmentations in plant and animal clades, suggesting a shared pattern [34,35]. By focusing on an ARF restricted plant clade of the Annonaceae family, we seek, in this study, to infer how many vicariance events occurred, if these are congruent with known aridification periods and if these events coincide with other independently estimated events in other ARF clades. In this context, studying speciation events through time in endemic ARF clades spanning both separated regions will provide complementary data to test these hypotheses. In order to test these hypotheses and infer vicariance ages, we can use dated molecular phylogenies of clades spanning east and west/central Africa, with endemic species in both regions [9,36].

Annonaceae are a pantropical family of trees, shrubs and lianas that are the most species-rich family of Magnoliales [37,38] and represent an important component of ARF in general [5] and locally [39]. Like numerous clades, Annonaceae are less diverse in Africa compared to the other main tropical rainforests regions, with 1100 species in South East Asia, 930 in the Neotropics and only 400 in Africa. The phylogeny of this family was studied for the first time more than 20 years ago [40], and since then, phylogenetic reconstructions have mainly been based on chloroplast markers [37,38,41,42]. More recently, the first Annonaceae phylogeny using hundreds of nuclear markers from a tailored baiting kit was published by Couvreur et al. (2019) [43], leading to robust phylogenetic inferences at generic and species levels. Such a kit provides a unique opportunity to infer phylogenetic relationships across Annonaceae and enables molecular dating within a single integrated framework.

The tribe Piptostigmateae (Malmeoideae subfamily) consists of six genera (*Brieya*, *Greenwayodendron*, *Mwasumbia*, *Piptostigma*, *Polyceratocarpus* and *Sirdavidia*), containing 31 species [43–49]. This tribe, together with its closely related tribe Annickieae (one genus/eight species [43]), presents an interesting biogeographic pattern, useful for testing the above hypotheses. Three genera contain endemic species in both ARF regions: *Greenwayodendron* (five GC/one EA endemics), *Annickia* (seven GC/one EA endemics) and *Polyceratocarpus* (six GC/two EA endemics). The Piptostigmateae tribe also contains two endemic sister genera with narrow ranges: *Sirdavidia* (GC), from the rainforests of Gabon and Cameroon, and *Mwasumbia* (EA), endemic to the coastal forests of Tanzania [43]. Estimating speciation events within this tribe between east and west/central African endemics and correlating these with known abiotic events responsible for rainforest fragmentation (i.e., aridification events) can improve our understanding of the ARF flora diversification.

Here, we aim to answer three main questions about ARF evolution using the Piptostigmateae and Annickieae tribes: (1) What were the diversification patterns driving Piptostigmateae evolution during the Cenozoic? (2) Has extinction played an important role in shaping current ARF diversity? (3) How many vicariance events do we infer within this clade and are these events congruent with known climatic and geological events?

## 2. Material and Methods

### 2.1. Taxon and Nuclear Marker Sampling

We sequenced one individual per species for the 23 available species, out of the 31 species known in Piptostigmateae (74%), covering all six genera of the tribe, and seven species out of eight, for the genus *Annickia* (Annickieae). We made sure to choose species from both ARF regions in the genera

*Greenwayodendron* and *Annickia* (endemic EA *Polyceratocarpus* species were not available). Our ingroup includes, in total, three endemic species from the East African region, and 27 endemic species from the Guineo–Congolian region (Table S1). Finally, we selected and added 23 species as outgroups from four different subfamilies: eight Malmeoideae (same subfamily as Piptostigmateae), 11 Annonoideae, three Ambaviodeae and one Anaxagoreoideae.

The Annonaceae bait kit [43] was used for DNA sequencing. This kit targets a total of 469 exons Annonaceae wide. Details on all 53 species (Table S2) used in this study, such as field sampling, DNA extraction and DNA sequencing are presented in Couvreur et al. (2019) [43]. We therefore retrieved fastq files from the AFRODYN bioproject (https://www.ncbi.nlm.nih.gov/bioproject/PRJNA508895), which contains demultiplexed, trimmed and sorted DNA paired reads (R1 forward + R2 reverse, 150 bp per read) of more than 120 Annonaceae species. Couvreur et al. (2019) [42] also provided the reference file for the baited regions, a fasta file comprising 469 targeted exons.

## 2.2. Contig Assembly, Alignment and Paralog Identification

Raw reads from 53 individuals were cleaned as in Couvreur et al. (2019) [43]. We processed our data with HybPiper (v. 1.2) [50] taking, as inputs, the clean fastq reads and the reference file containing the targeted exon sequence data. We retrieved contigs of exons and flanking introns and we aligned them separately with MAFFT (v. 7.305) [51]. Poorly aligned regions were cleaned using Gblocks (v. 0.91b, gaps allowed) [52]. HybPiper flags potential paralogs, which we assessed and removed from downstream analyses as follows. If several contigs covered more than 85% of the length of the same target gene, they were set aside for verification. Phylogenetic trees were created for each group of potential paralogs using RAxML [53]. It might be expected that alternative sequences were grouped by individual (e.g., different alleles in the same individual). However, if sequences corresponding to the "main" contig (i.e., the contig best matching the target reference) and other alternative sequences (numbered from 1 to n) clustered into distinct groups in the tree, these groups were considered as paralogs. In this case, we removed the entire locus from subsequent analyses.

## 2.3. Exons and Introns Selection for Molecular Dating

To minimize missing data for our phylogenetic inference, we selected exons and introns that were present in more than 75% of individuals and for which we recovered at least 75% of the exon's length (referred to as the 75/75 dataset), following Couvreur et al. (2019) [43]. Firstly, a phylogenetic tree was inferred with RAxML [53] for each of the exons and introns in our 75/75 dataset. These trees were then rooted one by one with the most phylogenetically distant outgroup available. We used TreeShrink [54] with default parameters, to automatically identify and remove outliers in our trees The number and length of loci in our relatively large dataset made it computationally intractable to use all of the data for divergence time estimation. To identify a subset of genetic markers that would provide accurate, computationally feasible estimates of divergence time, we took a "gene-shopping" approach using the SortaDate pipeline [55]. Root-to-tip variances (that is the variance of the distance of branch lengths from root to each tip) were calculated for each tree and used as indicators to determine which trees are most likely to follow a strict molecular clock. This was done to reduce model complexity and select exons and introns with similar (but not identical) evolutionary rates. However, we did not apply the phylogenetic relationship filter, i.e., we did not select genes that produced a particular phylogenetic relationship. A total of 32 sequences (31 exons and 1 introns) with the lowest "root-to-tip" values were selected for the dating analysis. The ModelTest-NG tool [56] was then used to identify the best nucleotide substitution models for each of the 32 sequences, according to the Bayesian Information Criterion (BIC).

## 2.4. Molecular Dating

The phylogenetic tree was inferred and dated using BEAST (v. 2.5.2) [57], a Bayesian approach for generating time-calibrated phylogenetic trees. Pirie and Doyle (2012) [58] concluded that the oldest

reliable age estimate for the Annonaceae crown node is *Futhabanthus*, a fossil flower which dates back to 89 Myr. *Endressinia* provides the most recent common ancestor (MRCA) of Magnoliaceae and Annonaceae at 112 Myr. As there is no other reliable fossil for Annonaceae, and no evidence of Piptostigmateae fossils, we chose to constrain the Annonaceae crown node between those dates (89–112 Myr) and apply a uniform prior following [58]. In addition, we imposed two topological constraints based on previous phylogenetic analyses: *Anaxagorea crassipetala* as the sister to the rest of the family, and *Anaxagorea crassipetala* plus the three sampled species of Ambavioideae as the sisters to the rest of the family [37,43]. After using Nested Sampling [59] to select the best model (see Table S7), we chose an uncorrelated lognormal relaxed molecular clock model, as well as a Yule tree prior model (identical net diversification rate in all branches, no extinction). We ran each analysis for 20 million generations, sampling every 2000 generations and repeated the process three times to compare the consistency of results across runs. We aimed to reach an adequate Effective Sample Size (ESS) of our posterior distribution above 200 for all parameter estimates. The output of each chain was then analyzed using Tracer (v. 1.7.1) [60] to check for convergence between the three runs. LogCombiner (v. 2.5.2) [61] was used to combine converging runs into a single chain (a burnin of 20% for each analysis). TreeAnnotator (v. 2.5.2) [61] was then used to determine the Maximum Clade Credibility (MCC) tree, as well as the mean ages, 95% highest posterior density (HPD) interval and the posterior probability (PP) of each node. We used the results of this dating analysis to find vicariance events across ARF history.

## 2.5. Diversification Analyses

Diversification analyses were conducted on the Piptostigmateae tribe only, because Annickieae is sister to all Malmeoideae and the sampling within the rest of the tribe (besides Piptostigmateae) is highly incomplete. The 21 outgroups as well as the seven Annickia were thus pruned from the MCC tree using the "drop.tip" function of the package ape (v. 5.3) [62,63]. For each analysis, the number of missing taxa in the phylogeny was specified for each of the following three methods, as our data contained 74% of known Piptostigmatae species. In order to cross validate our results, we undertook three different diversification analyses under maximum likelihood and Bayesian approaches. We first used RPANDA (v. 1.5) [64], to fit different diversification models to the dated phylogenetic tree under a maximum likelihood framework and estimate speciation and extinction rates through time. After several tests to find the best starting parameters (a range of priors between 0.1 and one were tested, with negligible change to the results), we set 0.2 events per lineage per Myr for the speciation rate and 0.05 events per lineage per Myr for the extinction rate. We then chose to test six different diversification models: two null models (time constant birth model and time constant birth–death model), and four time-dependent models: (a) pure-birth (no extinction) exponential speciation rate, (b) birth–death with exponential speciation rate and constant extinction rate, (c) birth–death with constant speciation rate and exponential extinction rate, and (d) birth–death with exponential speciation and extinction rates. The choice of the best model was made using the corrected Akaike information criterion (AICc) following the recommendations of Burnham and Anderson (2002) [65].

Secondly, in order to conduct a comparison with RPANDA results, we used a Bayesian approach as implemented in TESS (v. 2.1.0) [66] to fit the same six diversification models as above and compute their marginal likelihoods. We set an exponential prior distribution with rate = 10 for the speciation and extinction rates for all models, and ran Markov chain Monte Carlo (MCMC) analyses (100,000 iterations, 10,000 burn-in). We then used steppingstone simulations to estimate the marginal likelihood for each of our models (100 stepping-stone iterations, 10 burnin). MCMC convergence was verified with the coda package to ensure that ESS values were at least above 100 (we also verified mixing with a visualization of trace and density of the MCMC), and models were compared using Bayes Factors (BF).

Thirdly, we used BAMM (v. 2.5.0) [67,68] to detect possible diversification rate shifts along the branches of the tree. This Bayesian inference method uses a reversible jump Markov chain Monte Carlo (rjMCMC) approach to explore various models, detect speciation rate shifts over time and estimate

diversification parameters (i.e., speciation and extinction). Initial parameters were estimated with the R package BAMMtools (v. 2.1.6) [67] (expected number of shifts = 1), and the other priors were set to their default values. We ran a rjMCMC for 10 million generations (sampling every 1000 generations, 10% burnin) in order to ensure the convergence on four Metropolis-coupled MCMC chains (default values). Effective sample sizes for the likelihood and number of shifts were verified (ESS > 200) using the package coda [69]. BF were then calculated with the computeBayesFactor command in BAMMtools to find the best fitting-model.

Finally, to test the hypothesis that the Piptostigmateae tribe was impacted by one or several mass extinctions events during its evolutionary history, we used CoMET implemented in TESS [66]. This Bayesian statistical method computes the joint posterior distribution of speciation-rate shifts, extinction-rate shifts, mass-extinction events and estimate their values by running rjMCMC simulations over multiple episodically varying birth–death models. This method has also been suggested to perform better with small phylogenies, as in our case, when compared to other methods such as BDSKY [70]. We set the priors of mass extinction and speciation rate shifts to one as we had no a priori information. We ran the rjMCMC with a constraint of 500 ESS for each shift minimum, and we considered shifts as significant if 2 ln BF ≥ 6 [71].

## 3. Results

### 3.1. Phylogenomics

We recovered all the 469 exons present in the baiting kit with their corresponding introns. Our 75/75 dataset contained 330 exons and 329 introns. A total of 17 exons and 17 introns were flagged by HybPiper as potential paralogs, and of these, 32 were part of the 75/75 dataset. These were discarded, leaving a total of 298 exons and 297 introns for downstream phylogenetic analyses. After using our "gene shopping" approach, the 31 exons and one intron selected based on root-to-tip variance had a total length of 22,001 bp. The appropriate nucleotide substitution models selected for each exon and intron can be found in Table S3. Our three MCMC chains converged to similar values and the combined ESS values was above of 200 for all parameters and greater than 600 for posterior and likelihood values. The majority of nodes were highly supported in the resulting MCC tree (Figure 2), with all PPs over 0.96, and over 80% of nodes having posterior probability (PP) equal to 1.0. Annickieae and Piptostigmateae were recovered as monophyletic with maximum support (PP = 1.0). Finally, our results also support the placement of Annickieae as a sister to the rest of the Malmeoideae subfamily.

### 3.2. Molecular Dating and Divergence Times

The crown node of Annonaceae was dated at 91.85 Myr (95% HPD interval: 89.00–97.74 Myr). The age of the crown node of the Piptostigmateae tribe is dated to 33.35 Myr (95% HPD interval: 30.00–37.12 Myr). The crown ages of extant lineages for the six Piptostimateae genera and *Annickia* are presented in Table S4. Nodes corresponding to speciation events between endemic species of the EA region (in red in Figure 2) and endemic species of the GC region are dated as follows: divergence between *Annickia kummeriae* and the rest of the Annickieae is estimated at 5.36 Myr (95% HPD interval: 4.63–6.17 Myr). The divergence between *Greenwayodendron usambaricum* and *Greenwayodendron gabonicum*, *G. glabrum*, *G. litorale*, *G. suaveolens* is estimated at 5.79 Myr (95% HPD interval: 4.94–6.68 Myr). Finally, divergence between *Mwasumbia alba* and *Sirdavidia solannona* is estimated at 16.13 Myr (95% HPD interval: 13.89–18.18 Myr). All the nodes described here are well supported with a PP of 1.0.

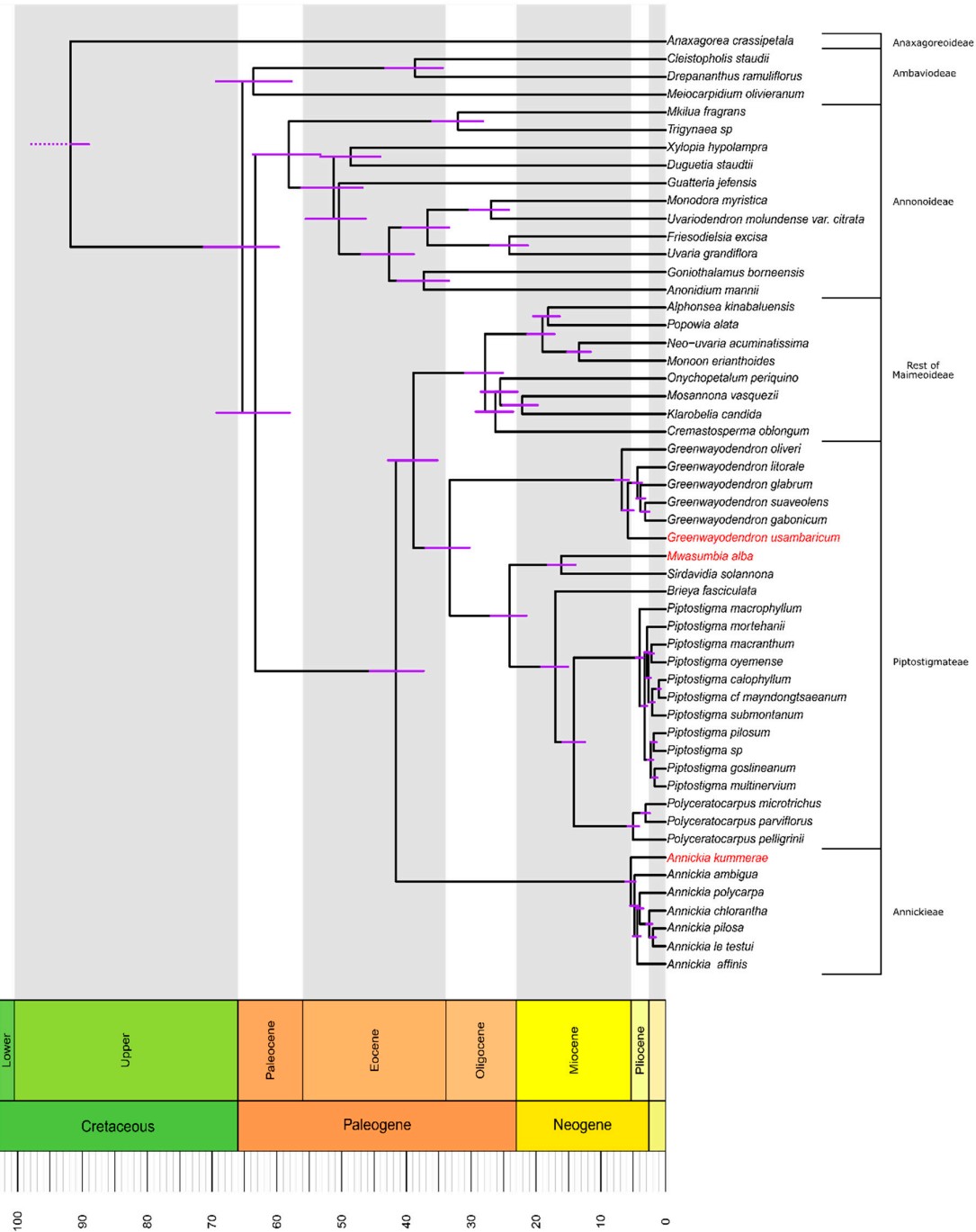

**Figure 2.** Maximum Clade Credibility (MCC) tree of the tribes Piptostigmateae and the Annickieae, based on 32 nuclear loci. Species colored in red are East African endemics. Purple bars represent the 95% highest posterior density (HPD) interval for each node. The last period corresponds to the Pleistocene epoch.

### 3.3. Diversification Analyses

Results of the diversification analyses with RPANDA supported the pure-birth (no extinction) exponential speciation rate as the best fitting model for Piptostigmateae, with speciation increasing through time (Table 1). Cross-validation with TESS also supported the pure-birth exponential speciation rate as the best fitting model (Table S5). However, Bayes Factor comparisons did not significantly distinguish this model from other models tested that had constant or no extinction (Table S5, difference in marginal likelihood with the simplest model (constant speciation/no extinction) < 2). Finally, the

BAMM analysis did not detect any significant rate shifts during the evolution of Piptostigmateae. ESS values for the likelihood and the number of shifts were greater than 2900, and BF analysis supported the null model (no shift) as the best fitting model (Table S6). This model indicates that speciation rates have gradually increased through time during the clades history (Figure 3), in agreement with the RPANDA and TESS results (Table 1, Table S5). It also shows a higher (but not significant) speciation rate for the *Piptostigma* genus (mean speciation rate through the clade = 0.2027) compared to others (*Polyceratocarpus* = 0.1845, *Greenwayodendron* = 0.1804, *Mwasumbia* = 0.1804, *Sirdavidia* = 0.1804, *Brieya* = 0.1804) (Figure 3). The CoMET analysis converged (ESS > 500) but we did not detect any significant mass extinction events (Figure 4E,F).

**Table 1.** Results from the RPANDA analyses. Abbreviations: speciation ($\lambda$); extinction ($\mu$); corrected Akaike Information Criterion (AICc); the difference in AICc between the model with the lowest AICc and the others ($\Delta$AICc); estimated speciation at present ($\lambda0$); rate of change in speciation rate ($\alpha$, from the present to the past, negative rate meaning speciation has increased through time); estimated extinction at present ($\mu0$); rate of change in speciation rate ($\beta$, from the present to the past).

| Models | Log Likelihood | AICc | ΔAICc | λ0 | α | μ0 | β |
|---|---|---|---|---|---|---|---|
| Exponential λ (no μ) | −64.9652 | 134.531 | 0 | 0.224 | −0.074 | - | - |
| Constant λ/constant μ | −66.0258 | 136.651 | 2.120 | 0.241 | - | 0.200 | - |
| Constant λ/exponential μ | −64.9436 | 137.150 | 2.619 | 0.198 | - | 0.057 | 0.008 |
| Exponential λ/constant μ | −64.9652 | 137.194 | 2.663 | 0.224 | −0.074 | $1.98 \times 10^{-7}$ | - |
| Constant λ (no μ) | −68.1224 | 138.435 | 3.904 | 0.127 | - | - | - |
| Exponential λ/exponential μ | −64.9652 | 140.153 | 5.622 | 0.224 | −0.074 | $2.20 \times 10^{-6}$ | 0.023 |

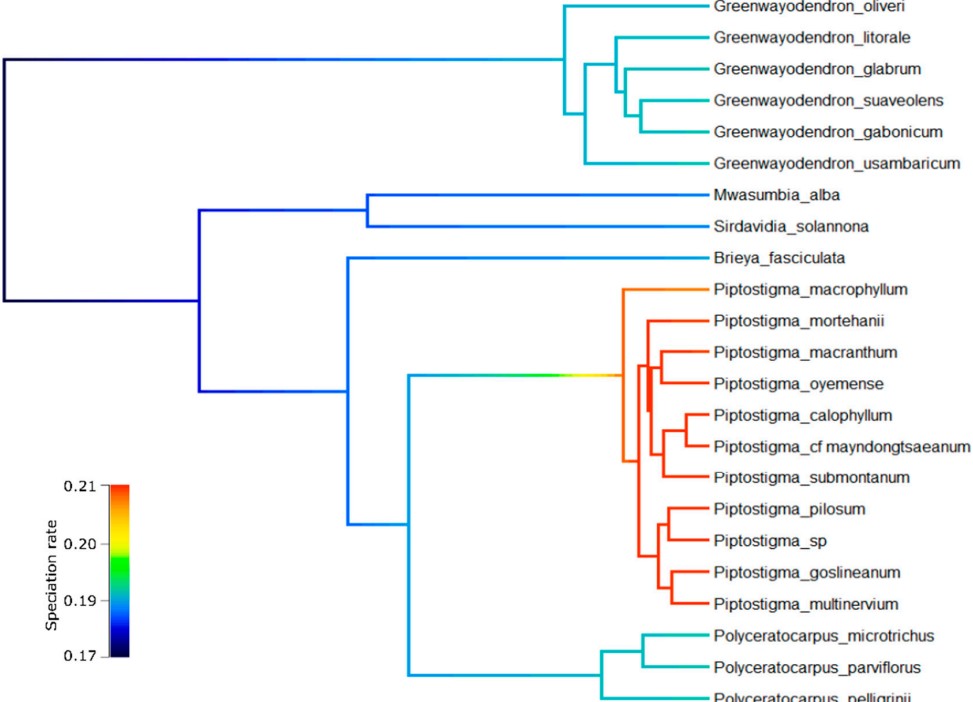

**Figure 3.** Results of the BAMM analysis. Phylogenetic tree of Piptostigmateae with inferred speciation rates on each branch (in events per lineage per million years).

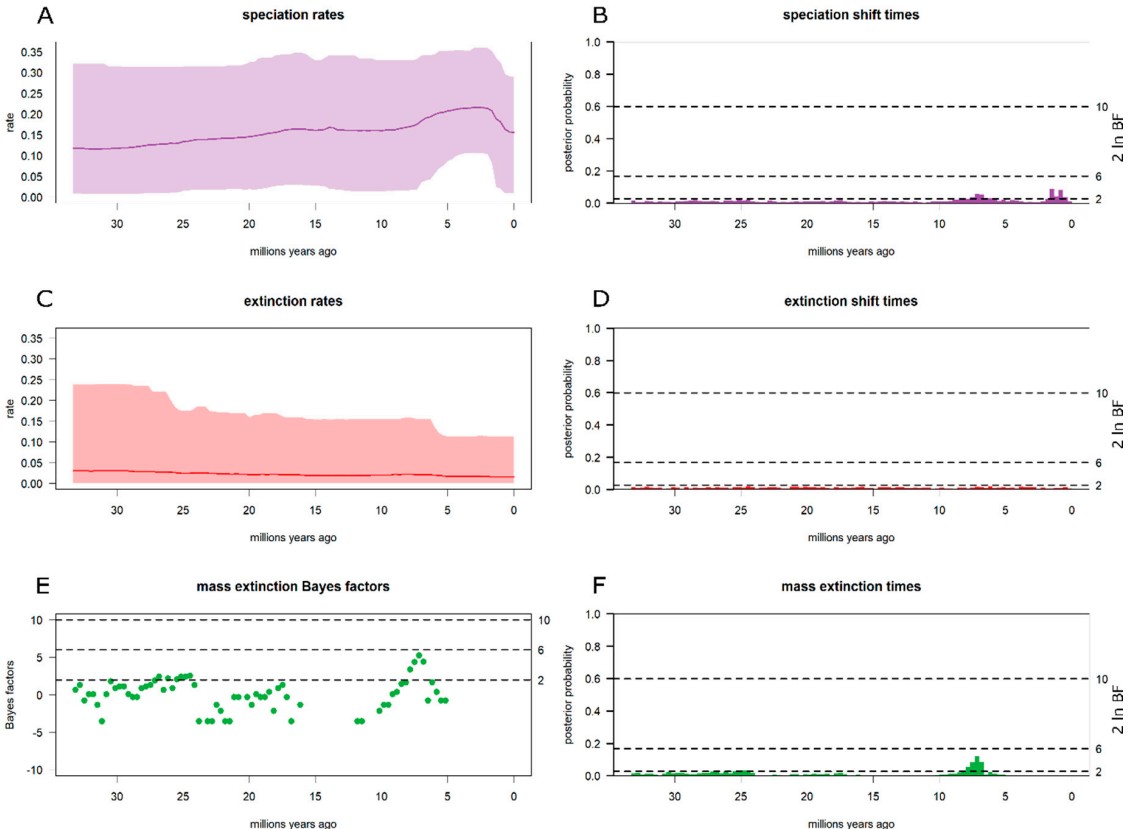

**Figure 4.** Results of the CoMET analyses. Inferred speciation (**A**) and extinction (**C**) rates through time (posterior mean and 95% credible interval), and possible mass extinction occurrence (**E**). Graph of the Bayes Factors (BF) testing models of speciation shift (**B**), extinction (**D**) and mass extinction (**F**) against null model. In total, 100 BF calculations were done from 33.35 Myr (Piptostigmateae crown node estimation) to present (≈1 BF estimation/0.33 Myr, represented by bars of posterior probability). A significant rate shift was indicated if 2 ln BF > 6.

## 4. Discussion

### 4.1. Temporal Estimates of African Rainforest Vicariance

In this study, we inferred a dated molecular phylogenetic tree of the Piptostigmateae and Annickieae tribes under a Bayesian inference method (BEAST, Figure 2). We used the same markers of Couvreur et al. (2019) [43], but we undertook a Bayesian phylogenetic analysis, which was not carried out in that paper (maximum likelihood only; a gene tree approach (ASTRAL) and concatenated approach using RAxML). Our Bayesian analysis, albeit based on 32 selected sequences, corroborated the results obtained using the gene trees species approach of Couvreur et al. (2019) [43] implemented in ASTRAL. Couvreur et al. (2019) [43] erected the Annickieae tribe based on the ASTRAL results, which is supported by our Bayesian analysis. Indeed, *Annickia* is inferred as a sister to the rest of the Malmeoideae with maximum support (Figure 2) (PP = 1).

Our dating analysis is the first time numerous nuclear markers were used to date clades within Annonaceae. Past studies have mainly relied on a few plastid markers [72,73] or full plastomes [74,75]. Interestingly, the 95% HPD interval of our molecular dating estimates of all the major clades of Annonaceae are consistent with previous global scale studies using BEAST but based on few plastid markers [41,76]. Nevertheless, the retrieved dates were overall more recent within the genus *Greenwayodendron* (Piptostigmateae) when compared to of the ages inferred by Migliore et al. [74] based on full plastome data and detailed intra-specific sampling. However, our age estimates do overlap based on the 95% HPD interval. We also note that, overall, our 95% HPD interval are smaller than the

abovementioned studies. This can be explained by the use of more sequence data (32 exons/intron versus 10 plastid markers), which might also be more informative. Our larger dataset allowed us to select the markers most appropriate for divergence time estimation, making our estimates the most accurate and reliable for the tribe to date.

Our study provides two different time estimations of speciation events between species of ARFs endemic to the GC and EA regions, all occurring during the Neogene (Miocene/Pliocene). The oldest inferred split is between the two genera Mwasumbia (endemic to EA) and Sirdavidia (endemic to GC) dated to the Late Early Miocene (mean: 16.13 Myr). This date falls into the Middle Miocene Climatic Optimum (MMCO) (17–14.7 Myr) [77] and coincides with the stabilization of the East Antarctic Ice Sheet [78]. More importantly, it is followed by the Middle Miocene Climatic Transition (MMCT) (approximately 14.2 to 13.8 Myr). The MMCT was characterized by a global cooling and an aridification of the Congo basin and East Africa caused by the uplift of the East African plateau as well as other factors such as the East Tethys sea final closure [79–86]. In addition to these intra African events, other factors such as the stabilization of East Antarctic Ice Sheet could also have led to the increased aridity in Africa [78]. The expansion of grassland ecosystems and rainforest contraction have been identified during this period [2,33,79,87] (reviewed by [88]). Moreover, multiple independent molecular dating studies have also suggested vicariance events between forest restricted clades during this period in plants [24–26,89] including other African Annonaceae tribes [9], and animals such as forest-restricted chameleons [35]. Therefore, the divergence between Mwasumbia and Sirdavidia seems likely to have been the consequence of a vicariance event due to the aridification of the African continent. Some studies have questioned this middle Miocene ARF reconnection [90,91]. Linder [30] argued that East African vegetation before the East African uplift was more likely to be woodland. This was further supported by the lack of fossil evidence of a pan-African rainforest during the early/mid-Miocene. Nevertheless, the fossil records for these regions remain poor [30]. However, the concordance in timing of our results with previous dated molecular studies (e.g., [9]) lends support to an Early Miocene reconnection of the African rainforest block followed by the mid-Miocene fragmentation.

The two other vicariance events are temporally congruent, dated to the end of Miocene/beginning of the Pliocene (mean ages: 5.36 and 5.79 Myr). This supports the idea that a single event affected the divergence of species in both Greenwayodendron and Annickia at this time. During the second half of the Miocene, drier climatic conditions led to the expansion of savannas and grasslands, with C4 photosynthetic plants becoming progressively dominant, notably in EA [2,30]. Studies also report a progressive diversification of animal clades living in arid and open ecosystems during the late Miocene in Africa [19,92,93]. Suitable conditions for ARF expansion arose again only during the Early Pliocene (5–3.5 Myr), where the palynological record points to a moist climate, rapid diversification of rainforest taxa and retraction of savannas [2,13,81,88,94]. Our molecular dating also revealed that the ages of crown nodes of each of the genera *Polyceratocarpus*, *Piptostigma*, *Annickia* and *Greenwayodendron* match this period, which corroborates the possible re-expansion of ARF at this time. There is little evidence suggesting that ARFs expanded continuously again from East to West during the second half of Miocene. Thus, these two speciation events resulting from the vicariance of a continuous ARF at 5 Myr appears unlikely. As we recover a very similar vicariance in the two distinct clades, we suggest that two hypotheses could explain this pattern. Firstly, these two speciation events are the result of a same vicariance event due to a pan-African forest fragmentation, but later in geological times. In fact, abrupt climate changes are documented during the Late Pliocene–Early Pleistocene, with drying and cooling phases once again allowing the spread of grasses on the continent and causing extinction of rainforest taxa [2,30,88]. As indicated before, climatic conditions of the Early Pliocene could have reconnected ARFs, so maybe the speciation events we infer here occurred during this mid-Pliocene transition. Second, these events, despite their similarities, might be due to independent dispersal events linked to bird or other animal dispersal events. Indeed, both *Annickia* and *Greenwayodendron* contain some species that are large trees reaching up to 45 m, while other Piptostigmateae genera generally do not exceed 20 m [44,45,47,49]. In addition, fruits (monocarps) in *Annickia* and *Greenwayodendron* are small,

clearly stalked and brightly colored appearing particularly adapted for dispersal by canopy dwelling birds or monkeys [47,49]. Finally, palynological data around 7 Myr indicate a possible forest expansion in both EA and GC [2,90], which could have reduced distances between rainforest blocks at this time, facilitating dispersal.

### 4.2. Piptostigmateae Diversification

Piptostigmateae started to diversify during the Paleogene around 36 Myr (Figure 2), just before the global and abrupt cooling event termed the Eocene–Oligocene Transition (EOT, 33 Myr). This event is suggested to have decreased tropical rainforest biodiversity worldwide [2,95] and is probably the cause of extinction in numerous plant groups, like African palms [96,97]. Palynological records show that the EOT led to an important turnover of biodiversity with the appearance of many current ARF clades during early Oligocene, such as the genus *Annona* [2]. However, the majority of Piptostigmateae species seem to have mainly diversified more recently, during the Pliocene, linked to improved climatic conditions for tropical rainforest taxa (see above) (Figure 2). Nonetheless, using BAMM, we did not detect significant speciation rate shifts at this time, which could be related to low statistical power linked to the low number of species known from the tribe and sampled here. Nevertheless, these results are in line with recent works on the Annonaceae family that found no evidence for diversification rate shifts for Piptostigmateae [98,99]. The BAMM analysis however also inferred a higher (but not significant) speciation rate for the genus *Piptostigma* (Figure 3). It remains unclear what might have led to this increase for this specific genus. *Piptostigma* is characterized within the tribe by the inner whorl of petals being much longer than the outer ones [46], a character only shared with the species poor genus *Brieya* (two species). All other genera in Piptostigmateae have longer outer petals than inner or are equal length [45]. Another distinctive character for *Piptostigma* is the presence of numerous secondary veins [46] with tight parallel tertiary venation (see Figure 1G) of the leaf blade (generally more than 20 pairs and up to 65 for some species, whereas other genera generally have less than 20 veins). High leaf vein density is suggested to increase photosynthetic and transpirational capacities [100,101] which could have conveyed a competitive advantage of *Piptostigma* over other Piptostigmateae genera, although this requires further testing.

Finally, because speciation within *Piptostigma* mainly took place during the Pleistocene (Figure 2), it cannot be excluded that diversification within this genus was linked to the lowland refuge hypothesis. This pattern, however, has rarely been reported in ARF restricted tree genera to date (e.g., *Carapa* [102]; *Coffea* [103]). In the latter genera, most species originated in the last 2 Myr. Most Pleistocene speciation events have been documented in herbaceous genera, such as *Begonias* or *Impatiens* (e.g., [7,104]), whereas tree genera have generally been inferred to originate before the Pleistocene [9,26,41,74,97,105,106].

The RPANDA analysis indicated that the speciation rate within Piptostigmateae increased exponentially through time, together with negligible extinction rates (Table 1). However, because we did not test any intermediate models between exponential and constant, the true speciation rate probably lines somewhere in between these two extremes. The negligible effect of extinction inferred on Piptostigmateae diversity is concordant with our BAMM analysis, which also found an increase of the speciation rate through time with no significant diversification rate shift (Figure 3). In addition, we did not detect any significant extinction shift, nor mass extinction events when using CoMET (Figure 4). Overall, our analyses do not detect a major role of extinction, either punctual or gradual, during the evolutionary history of Piptostigmateae. If interpreted as such, the low extinction rates inferred here are in line with a few other studies in the region. For example, in the palm subtribe Ancistrophyllinae, a clade of 22 climbing palm species (rattans) occurring across ARF, a near complete dated molecular phylogeny estimated low extinction rates [97]. They did, however, detect a signal of an ancient mass extinction event at the Oligocene–Eocene boundary, which was not detected here. In addition, a global study of palm diversification inferred overall low extinction rates for palms as a whole and did not detect a significant decrease in diversification rates for African genera [107]. These authors suggested that the "odd man out" pattern might not be the result of high extinction rates, but lower speciation

rates in Africa. Interestingly, this hypothesis has received additional support from other pantropical plant families such as Sapotaceae [108] or Chrysobalanaceae [109] in which higher speciation rates and not extinction rates have been detected outside of Africa. Finally, for plants as a whole, a recent study found both higher speciation and extinction rates for the Neotropics, suggesting that higher diversity of the Neotropics is linked to higher turnover of taxa through time [110], but see [111]. Thus, these studies do not support the hypothesis that lower African rainforest diversity is linked to higher extinction rates [2,17,18]. However, we are aware of the difficulties of inferring extinction from molecular phylogenies without fossil data ([112], but also see [113]) and thus our results on extinction should be considered with caution.

## 5. Conclusions

Our dated molecular tree of both Piptostigmateae and Annickieae tribes detected two major vicariance events between central African and East African rainforest lineages: one during the Mid-Miocene, possibly linked to climatic and/or geological changes fragmenting the pre-existing pan-African rainforest at this time; the second one is estimated to have occurred at the beginning of the Pliocene, and is harder to link to a rainforest vicariance event. Our results show that Piptostigmateae species mainly diversified during the Pliocene and Pleistocene. The genus *Piptostigma*, with 13 species, is one of the rare examples of tree clades diversifying mostly during the Pleistocene. We did not detect high levels of overall extinction and/or mass extinction events to explain present-day diversity. This suggests a possible low impact of extinction of the evolutionary history of this clade. Therefore, our results are in line with other studies detecting low extinction rates within some African clades. In addition, Piptostigmateae was not associated with significant shifts in diversification rates. Indeed, the best fitting model suggested that the speciation rate of Piptostigmateae increased exponentially over time. These results imply that the global diversity of this tribe was not impacted by major abiotic changes during the Neogene. Overall, our study adds to the knowledge of ARF evolution and the patterns of diversification that have occurred in this species-rich ecosystem.

**Supplementary Materials:** The following are available online at http://www.mdpi.com/1424-2818/12/6/227/s1, Table S1: List and distribution of the 30 species used in this study (23 from the Piptostigmateae and 7 from the Annickieae) used for the phylogenetic reconstruction, with their main floristic bioregions as identified by Droissart et al. (2018) [114]. Grey highlights strict East African endemic species. All the others are endemics of the Guineo–Congolian region. Table S2: Specimen information used for this study with associated sequencing statistics. Total reads: Total number of reads recovered for each barcoded tag; mapped reads: number of reads mapped to reference exonic sequences; % enrichment: percentage of reads correctly mapped to reference exonic sequences. 10x coverage: proportion of the reference sequences covered with at least 10 base pairs; mean depth: average number of reads covering reference sequences. Table S3: Table listing the 32 selected sequences chosen for the BEAST analysis, with their best nucleotide substitution models selected with the BIC criterion and their tree length with root-to-tip variance. Table S4: Age estimate of the extant taxa for the six Piptostigmateae genera and Annickia using BEAST. The crown node estimation of Brieya was not possible because we did not include the second species of this genus. Table S5: Results of the TESS analysis. ESS for all MCMC where above 300 except for the Exponential λ/exponential μ model (less than 100). Abbreviations: λ: speciation; μ: extinction; λ0: estimated speciation at present; α: rate of change in speciation; μ0: estimated extinction at present; β: rate of change in speciation rate (from the present to the past). Table S6: Output of the computeBayesFactors command from BAMMtools. Bayes Factors values for the model with k shifts relative to the model with 0 rate shifts. Table S7: Results of the Nested Sampling model selection.

**Author Contributions:** Conceptualization, T.L.P.C. and A.J.H.; methodology, A.J.H., K.B.; validation, K.B., A.J.H. and T.L.P.C.; formal analysis, B.B. and A.J.H.; resources, T.L.P.C. and J.-P.G.; data curation, J.-P.G. and K.B.; writing—original draft preparation, B.B. and T.L.P.C.; writing—review and editing, A.J.H., B.S. and J.-P.G.; visualization, B.B. and T.L.P.C.; supervision, B.S. and T.L.P.C.; project administration, T.L.P.C.; funding acquisition, T.L.P.C. All authors have read and agreed to the published version of the manuscript.

**Funding:** This research received no external funding.

**Acknowledgments:** We thank Raoul Niangadouma and Narcisse Kamdem for help in the field. The authors acknowledge the IRD itrop HPC (South Green Platform) at IRD Montpellier for providing HPC resources. This study was supported by the Agence Nationale de la Recherche (grant number ANR-15-CE02-0002-01 to TLPC). We are grateful to the Centre National de la Recherche Scientifique et Technique (CENAREST), the Agence National des Parcs Nationaux (ANPN) and Prof. Bourobou Bourobou for research permits (AR0020/16; AR0036/15

(CENAREST) and AE16014 (ANPN)). Fieldwork in Cameroon was undertaken under the "accord cadre de cooperation" between the IRD and Ministère de la Recherche Scientifique et Technique (MINRESI). Finally, we thank three anonymous reviewers for providing detailed comments on a previous version of this article.

**Conflicts of Interest:** The authors declare no conflict of interest.

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
