# Peer review of "Diversification of African Rainforest Restricted Clades: Piptostigmateae and Annickieae (Annonaceae)"

_diversity, doi:10.3390/d12060227_

Round 1
Reviewer 1 Report
Excellent job with the reviews. I just have but a few suggestions in the attached document. Looking forward to the final version.

Author Response
As all the new comments were typo or formulations, all the changes have been made directly in the corrected manuscript.

Reviewer 2 Report
The authors have revised the paper appropriately according to comments, but there seem to be still some points that authors should answer and revise. Please, see my specific comments below.
>Furthermore, although author stated lower diversity in African rain forests, compared to
>those of Neotropics and South East Asia in Introduction Part, there was insufficient statement
>about the reason of the difference of the diversity in Discussion Part. I also expect the revision
>including what authors think about this difference.
>Indeed, we do not discuss the odd man out in detail here. We didn’t want to talk too much about
>this because to really test this pattern a pan tropical sampling is needed. However, we feel that
>our result provides some limited addition to the discussion. Thus we mention it briefly here as
>not to over emphasize our results given the intra Africa scope of our study.
If so (but for me, this is unsatisfactory for the revision), the first keyword of your study is not the word or hypothesis of odd man out. There seems to be a conflict between the story of the manuscript and keyword.
> l.30; I wonder the former three keywords are appropriate for the critical words
>standing for this manuscript.
>We added the word hypothesis.
How about the latter two? According to your represented results, there could be other optimal candidates for keywords, rather than the current ones. You also describe “results of extinction should be considered with caution” in l.426.
l.70-71; Previous sentence is no need
l.116: The usage “evolution” is strange (= evolution of forest ?). Instead, it can be recommended to use “diversification of ARF flora” here.
>l.143-146; The applied “gene shopping” approach should be explained more. The
>explanations of the approach and the significance/validity are unclear.
>We have added additional explanation of the gene shopping approach (Line 159)
I see your revision, and then, “root-to-tip variance” seems to be the index that you should explain a little bit more. What’s the meaning of this (variance of what)? What is the difference from the alternative index nucleotide diversity (π). Please add some explanation to both around l.163 and Table S3.
Figure 2; Check the border line between “Rest of …” and “Piptostigmateae” again.
l.261; The sentence of the revised version inserted “0” seems strange for me. What’s the meaning of “0”?
Table 2; You moved previous Table 1 to Table S4, hence you must change Table 2 into Table 1 in the revised version.
>Table S2; Please modify the order of samples according to the Table S1 (for easy
>understandings). What are the meanings of values in “number”, “INDEX”, “10x
>coverage”, and “maen depth” in Table. For later two, why authors used “,” (instead of “.”)?
>We remove the “INDEX” and “TAG” column as there are not relevant to this study.
>We explain the 10x coverage ane mean depth in the title of the table.
I see the revised version, but I still ask you for the unclear meaning of “number”.
Abbreviation must be defined where you used at first, while your revised version is not; ESS in l.217, PP in l.250, HPD in l.259.
Author Response
In italics our answers to round one (A1), in bold our answers to round 2 (A2).
Reviewer 2 new comments
R1: Furthermore, although author stated lower diversity in African rain forests, compared to those of Neotropics and South East Asia in Introduction Part, there was insufficient statement about the reason of the difference of the diversity in Discussion Part. I also expect the revision including what authors think about this difference.
A1: Indeed, we do not discuss the odd man out in detail here. We didn’t want to talk too much about this because to really test this pattern a pan tropical sampling is needed. However, we feel that our result provides some limited addition to the discussion. Thus we mention it briefly here as not to over emphasize our results given the intra Africa scope of our study.
R1: If so (but for me, this is unsatisfactory for the revision), the first keyword of your study is not the word or hypothesis of odd man out. There seems to be a conflict between the story of the manuscript and keyword.
A2: We have removed this as a keyword. I guess we can say that out the study is within the context of the “odd man out” pattern but doesn’t explicitly aims to test it, which would require a larger dataset inter continent.
R1: l.30; I wonder the former three keywords are appropriate for the critical words standing for this manuscript.
A1: We added the word hypothesis.
R1: How about the latter two? According to your represented results, there could be other optimal candidates for keywords, rather than the current ones. You also describe “results of extinction should be considered with caution” in l.426.
A2: We have added new keywords better suited to our study as suggested.
R1: l.70-71; Previous sentence is no need
A2: We removed it.
R1: l.116: The usage “evolution” is strange (= evolution of forest ?). Instead, it can be recommended to use “diversification of ARF flora” here.
A2: We changed the end of the sentence by : “can improve our understanding of the ARF flora diversification”.
R1: l.143-146; The applied “gene shopping” approach should be explained more. The explanations of the approach and the significance/validity are unclear.
A1: We have added additional explanation of the gene shopping approach (Line 159)
R1: I see your revision, and then, “root-to-tip variance” seems to be the index that you should explain a little bit more. What’s the meaning of this (variance of what)? What is the difference from the alternative index nucleotide diversity (π). Please add some explanation to both around l.163 and Table S3.
A2: We have added a definition of the root to tip variance:
Line 163: “(that is the variance of the distance of branch lengths from root to each tip)”
R1: Figure 2; Check the border line between “Rest of …” and “Piptostigmateae” again.
A2: We move it to it right place, thank you.
R1: l.261; The sentence of the revised version inserted “0” seems strange for me. What’s the meaning of “0”?
A2: Sorry, it was a typo, we removed it.
R1: Table 2; You moved previous Table 1 to Table S4, hence you must change Table 2 into Table 1 in the revised version.
A2: Done.
R1: Table S2; Please modify the order of samples according to the Table S1 (for easy understandings). What are the meanings of values in “number”, “INDEX”, “10x coverage”, and “maen depth” in Table. For later two, why authors used “,” (instead of “.”)?
A1: We remove the “INDEX” and “TAG” column as there are not relevant to this study.
We explain the 10x coverage ane mean depth in the title of the table.
R1: I see the revised version, but I still ask you for the unclear meaning of “number”.
A2: The “number” corresponds to the collector’s number, a standard botanical practice (collector name and number represents the unique identification of a collection needed to find the specimen in order to check its identification if needed). We have added “collector number” to be more explicit.
R1: Abbreviation must be defined where you used at first, while your revised version is not; ESS in l.217, PP in l.250, HPD in l.259.
A2: ESS is already defined before in the text (ESS l.187, PP l.193, HPD l.192).
Reviewer 3 Report
The authors have addressed in this revised version any and all concerns I had with the original version.
Author Response
Reviewer 3 didn’t address any new comments.
This manuscript is a resubmission of an earlier submission. The following is a list of the peer review reports and author responses from that submission.
Round 1
Reviewer 1 Report
This phylogenomic study uses clades Piptostigmateae and Annickieae (Annonaceae) to address African rainforest fragmentation/connectivity through time and its impact in tree species diversification rates. To do so, the authors estimate divergence times and explore a number of diversification scenarios with a number of existing tools. Though this is a fine study, some work is needed to strengthen its conclusions. I see five main weaknesses. First, your hypotheses need a redo. Rather than “one” vs. “many” fragmentations, I would test “extinction” vs. “speciation” shaping present-day diversity, in the clades of interest in light of these fragmentations. Second, it would be best to rerun HybPiper with version 1.3.1 and split supercontigs into exons on one hand and flanking introns on the other and then redo SortaDate (please, clearly state your parameterization). Third, rerun BEAST and test different models (e.g., strict vs. relaxed clocks or yule vs. birth-death processes) and estimate MLEs using PS or SS to choose the best-fit model. Fourth, explore intermediate scenarios where speciation is somewhere in between constant and exponential and try a BDSKY approach (as the one proposed by Culshaw et al. 2019) capable of dealing with small phylogenies, such as yours. Fifth, be more cautious presenting your results in light of the limitations of the methods here implemented. Detailed suggestions follow.
Lines 5 and 6: please, change semi-colons for commas.
INTRODUCTION
Please, change ARF to ARFs if plural (e.g., lines 12, 24, 27, 34, 39, 41, 43, and so on) and keep ARF when singular (e.g., 36, 52, and so on) throughout the document. Thank you.
Line 20: please, remove comma after rates.
Line 36: please, change "when the ARF floral diversified have been" to "when the ARF flora diversified has been".
Lines 52–58: I am missing mention to well-established barriers to gene flow in Guinean forests such as the Dahomey Gap (please, see comments to Figure 1 below). Similarly, in Eastern Africa, montane tropical forests are very different from lowland tropical forests.
Line 63: please, change "that ARF have stretched" to "that ARFs could have stretched".
Figure 1: there is a known N-S barrier to gene flow in lower Guinea (Heuertz et al. 2014, PLoS ONE; Ley et al. 2014, Front. Genet.; Duminil et al. 2015, J. Biogeogr.; your very own Helmstetter et al. 2020, bioRxiv), which could be linked to the onset and cessation of seasonal rainfall (Dunning et al. 2016, J. Geophys. Res. Atmos.). Could you, maybe, show rainfall regime shifts in your map (as a line dotted line or something along those lines)? Otherwise, a line representing this observed N-S barrier would be much appreciated. Similarly, with regards to Upper Guinea, I would really appreciate it if you could highlight the Dahomey Gap (again, this is a known barrier to gene flow). Likewise, Eastern African montane forests are quite different from coastal ones. I would appreciate some sort of division be shown in this figure. Thank you.
Line 69: please, remove "on" at the beginning of this line so that it reads "fragmented only once".
Lines 68–72: If I understand correctly your hypotheses are "one" vs. "many" forest fragmentations. I am pretty sure the literature has agreed on "many" for a long while now. Also, Davis et al. 2002 (your reference # 32) does not state a single forest fragmentation took place but, rather, that their divergence time estimation is congruent with the onset of aridification in Africa. Those are two different things. Same goes for Axelrod & Raven 1978 (your reference # 17). They explain how successive aridification events decimated a once widespread Paleogene flora (a succession of many fragmentations through time since the onset of aridification processes). Again, this does not mean fragmentation occurred just once. On another hand, Burguess & Clarke 1998 (reference # 23) mainly focus on coastal Eastern African forests (forests that I would like you to mention in your introduction, please). In other words, I don’t see much support for the "one-time-only" hypothesis (for sure nothing published in recent times). Personally, rather than ask one vs. many, I would ask, how many and what impact did it have in, e.g., Piptostigmateae (Malmeoideae, Annonaceae). However, I think it would have been better to ask whether extinction or speciation shaped the diversity of the aforementioned clade or a combination of both.
Line 76: Rather than “Other molecular dating studies”, I would say “Most other molecular dating studies”.
Line 86: please, change “markers based a tailored” to “markers from a tailored”, please.
Lines 89–90: please, remove space between frame and work so that it reads as “framework”.
Line 93: “provide an” reads weird, maybe change to “present an”.
Lines 106–107: as previously mentioned, rather than one versus many I would ask how many.
Material and Methods
Line 116: why so many taxa in the outgroup?
Line 129: why HybPiper 1.2? Version 1.3.1 has been around for over a year now and has greater capabilities as well as takes care of a number of bugs. It really doesn’t take that long to rerun this pipeline.
Lines 131–132: Other than Gblocks (please change spelling to match this one), why not use other tools, such as TreeShrink (Mai & Mirarab 2018, BMC Genomics), to automatically identify and remove possible outliers?
Lines 132–137: wording in this sentence is confusing. Did you or did you not use the approach here described to remove potential paralogs from downstream analyses? Please, rephrase to remove ambiguity, thank you.
Lines 143–147: what thresholds did you use when applying SortaDate to your 75/75 data matrix? Also, why not separate introns from exons? That is, divide supercontigs into exons, on one hand, versus introns, on the other (since nucleotide substitution models are probably very different between these two) and rerun ModelTest-NG on them.
Line 157: why use a uniform prior? Uniform priors are quite biased as they give equal importance to all values for a given interval. Speaking of which, what interval are you exploring? What’s your mean? And your standard deviation? A normal prior is less biased when imposing this type of constraint. Why not use a secondary calibration on your root node and other maximally supported nodes (away from the tips)? E.g. those corresponding to topological constraints (lines 157–159).
Lines 159–161: if you do model selection for each of your supercontigs (please, divide exons from introns and rerun ModelTest-NG), why aren’t you doing any model selection here? You have used SortaDate to select genes likely evolving under a strict clock and then you run analysis under a relaxed molecular clock model (by the way, what distribution do you impose on this relaxed clock? Is it lognormal by any chance?). I’m confused. You should test different clock and tree models (other than under a pure birth model—that is, a Yule process—, one can also model speciation under a birth-death process). If your time-calibrated phylogeny does not contemplate extinction (Yule process), I am not surprised you aren’t finding any later on. You can easily use path sampling (PS) and Stepping-stone Sampling (SS) to estimate marginal likelihoods (MLEs) and select the best supported model (which could be a strict clock model and a BD speciation process instead of what you have arbitrarily chosen; and if it isn’t arbitrary, please, properly justify it).
Lines 163–166: I am surprised you consider ESS > 100 adequate. By default, Tracer has the threshold set at 200. Though, as long as mixing is acceptable, this shouldn’t be an issue. However, you do not explicitly state whether mixing is any good. Is mixing any good?
Line 167 (and line 229): please, change “Maximum Clade Credibility Tree (MCCT)” to “Maximum Clade Credibility (MCC) tree”, which is the convention in the existing literature (see any paper by the developers of BEAST), and MCCT to MCC tree in all subsequent mentions of this acronym (e.g., lines 174 and 224), thank you.
Line 168: please, change “density (HPD) and” to “density (HPD) interval and”.
Lines 172–174: this is why I do not understand why you have such a large outgroup. Please, explain. Also, why not try a clade dependent diversification approach such as MEDUSA (available in R package Geiger) that doesn’t require complete sampling?
Line 181: could you have some sort of supplementary table summarizing parameter exploration, please?
Lines 182–187: this is why I do not understand why no model testing is done for divergence time estimation. I feel your tests are circular. When inferring divergence times, you impose a Yule process and then you get pure birth with no extinction as your most optimal result. Also, why exponential rates? Are there no other options?
Line 194: You are clearly familiar with Stepping-stone Sampling (SS) to estimate MLEs for model selection (see comment to lines 159–161), why not use it to select the best fit model for divergence time estimation?
Lines 195 and 205: I find 100 a suboptimal ESS threshold. Could you comment on the mixing?
Line 205: how were Bayes Factors calculated?
Lines 207–208: why not use the approach proposed by Culshaw et al. 2019 (doi: 10.3929/ethz-b-000348893), which relies on Bayesian birth-death skyline (BDSKY) models, to detect mass extinction events (instead of TESS)? This BDSKY approach performs better than CoMET for smaller phylogenies and it is available in BEAST2.
RESULTS
Line 382: please, change “Highest Posterior Density (HPD)” for “HPD interval”. You already explain this abbreviation elsewhere. No need to repeat it here.
Lines 240–241 and 248–250: I do not see the point of this table in the main text. Most of that information is already available in your chronogram and the text. Please, move it to supplementary materials or completely remove it.
Lines 252–255: how is this not circular? Yule chronogram, yule diversification rate.
Line 265, 281, and 391: please, change “COMET” to “CoMET”, thank you.
Lines 265–266: I understand CoMET struggles with small phylogenies, such as yours, and I urge you to try a BDSKY model in BEAST2 instead.
DISCUSSION
Lines 293–294: you need to separate exons from flanking partial introns (now together in supercontigs).
Lines 295–297: your “gene shopping approach” (SortaDate) picks “genes” most congruent with the species tree, so this isn’t surprising in the least. I do not understand why it is highlighted here. Maybe change wording to reflect this expectation, given the chosen approach for divergence time estimation.
Lines 305–306: please, note that your choice of priors could also lead to an increase in precision.
Line 310: this date coincides with the stabilization of the East Antarctic Ice Sheet (EAIS), see Flower & Kennett 1994.
Line 311: change “Ma) and followed shortly after the Middle Miocene Climatic Optimum (MMCO)” to “Ma), which followed the Mid Miocene Climatic Optimum (MMCO)”.
Lines 313–314: in addition to the uplift of the African Plateau and the final closure of the Tethys sea, distant geological events, such as the previously mentioned stabilization of the EAIS, also played a role in the aridification of the Congo Basin (and no doubt in the global cooling mentioned in line 312).
Lines 315–318: in addition to Dimitrov et al. 2012, Pokorny et al. 2015, Tosso et al. 2018, and Makga et al. 2020, see studies on legume subfamily Detarioideae by de la Estrella et al. 2017 (doi: 10.1111/nph.14523) and Choo et al. 2020 (doi: 10.1016/j.ympev.2020.106752), for further evidence.
Line 336: I would rephrase “Greenwayodendron were around this period” so that it reads “Greenwayodendron match this period” instead.
Line 340: change “speciation date in” to “speciation timing in”, please.
Line 345: In addition to Morley 2000, Linder 2017, and Jacobs et al. 2010, I would also cite Sepulchre et al. 2006 and Senut et al. 2009 in here.
Line 354: remove space after “time” and before comma, please.
Lines 365–367: which is why I am recommending the approach proposed by Culshaw et al. 2019.
Line 368: a MEDUSA-like approach could maybe pick up this clade-specific shift.
Line 382–383: See Piñeiro et al. 2019 (doi: 10.1101/811463) for a recent study on refugia at the population level (as well as your own Helmstetter et al. 2020, doi: 10.1101/807727).
Line 386–388: regarding your RPANDA analyses, maybe speciation is neither constant nor exponential, but you are not contemplating any intermediates between these two extremes (and they are quite the extremes, if I may say). Thus, I would not go as far as to say speciation is exponential.
Lines 395–397: see also the aforementioned de la Estrella et al. 2017.
Lines 399–403: in Pokorny et al 2015 one can see that speciation and extinction are clade dependent (see figure 5 in that paper). So, neither high extinction nor high speciation but probably a number of different scenarios depending on the clade at hand and its tolerance to aridification-tropicalization shifts. The picture is a lot more complex than you are painting it to be.
Lines 405–406: but see de la Estrella et al. 2017.
Lines 419: please, change “low extinction rate within African clades” to “low extinction rate within some African clades”.
Line 415: see Choo et al. 2020 for an example in legume genus Daniellia.
Lines 420–422: I repeat, you are not testing any intermediate options between constant and exponential speciation.
Author Response
Point by point reply to the Review Report (in bold)
Please, change ARF to ARFs if plural (e.g., lines 12, 24, 27, 34, 39, 41, 43, and so on) and keep ARF when singular (e.g., 36, 52, and so on) throughout the document. Thank you.
We have changed as suggested
Line 20: please, remove comma after rates.
Done
Line 36: please, change "when the ARF floral diversified have been" to "when the ARF flora diversified has been".
Done
Lines 52–58: I am missing mention to well-established barriers to gene flow in Guinean forests such as the Dahomey Gap (please, see comments to Figure 1 below). Similarly, in Eastern Africa, montane tropical forests are very different from lowland tropical forests.
Today, the Dahomey gap is certainly a possible gene flow barrier to TRF plant species. In this sentence, we are referring to older speciation events not relevant to the age of the Dahomey gap which, from what we know, has “flipped” between TRF and savanna over the past 7 million years, with most vicariant events occurring during the last 3-2 million years, mainly documented in animals. The importance of the Dahomey gap in above species level speciation is not established yet. Even limits of the upper and lower guinea at floristic levels are disputed based on what data is used (family-specific studies, whole vegetation, trees versus shrubs). Finally, our study is focused on East and West/central events. For these reasons we do not mention the role of the Dahomey gap in our MS, because it would be fairly long to explain and off topic.
Line 63: please, change "that ARF have stretched" to "that ARFs could have stretched".
Done
Figure 1: there is a known N-S barrier to gene flow in lower Guinea (Heuertz et al. 2014, PLoS ONE; Ley et al. 2014, Front. Genet.; Duminil et al. 2015, J. Biogeogr.; your very own Helmstetter et al. 2020, bioRxiv), which could be linked to the onset and cessation of seasonal rainfall (Dunning et al. 2016, J. Geophys. Res. Atmos.). Could you, maybe, show rainfall regime shifts in your map (as a line dotted line or something along those lines)? Otherwise, a line representing this observed N-S barrier would be much appreciated. Similarly, with regards to Upper Guinea, I would really appreciate it if you could highlight the Dahomey Gap (again, this is a known barrier to gene flow). Likewise, Eastern African montane forests are quite different from coastal ones. I would appreciate some sort of division be shown in this figure. Thank you.
The reviewer here is referring to gene flow barriers, which are a within-species limit. This MS is not able to infer patterns in intraspecific structuring but speciation events. Thus we do not think it is relevant to add the suggested barriers to our figure. By diversification, we mean species-level diversification, and not within-species diversification as it is sometimes employed in phylogeographic studies.
The map represents the distribution of forested regions across tropical Africa. We agree with the reviewer that this might be too general within the scope of our study. We have thus changed this map to depict only tropical rain forests (lowland and premontane), taken from Couvreur et al. 2008. This map also clearly shows the breaks in TRF cover across the region, including the Dahomey gap.
Line 69: please, remove "on" at the beginning of this line so that it reads "fragmented only once".
Done
Lines 68–72: If I understand correctly your hypotheses are "one" vs. "many" forest fragmentations. I am pretty sure the literature has agreed on "many" for a long while now. Also, Davis et al. 2002 (your reference # 32) does not state a single forest fragmentation took place but, rather, that their divergence time estimation is congruent with the onset of aridification in Africa. Those are two different things. Same goes for Axelrod & Raven 1978 (your reference # 17). They explain how successive aridification events decimated a once widespread Paleogene flora (a succession of many fragmentations through time since the onset of aridification processes). Again, this does not mean fragmentation occurred just once. On another hand, Burguess & Clarke 1998 (reference # 23) mainly focus on coastal Eastern African forests (forests that I would like you to mention in your introduction, please).
[cited in the intro]
In other words, I don’t see much support for the "one-time-only" hypothesis (for sure nothing published in recent times). Personally, rather than ask one vs. many, I would ask, how many and what impact did it have in, e.g., Piptostigmateae (Malmeoideae, Annonaceae). However, I think it would have been better to ask whether extinction or speciation shaped the diversity of the aforementioned clade or a combination of both.
We appreciate the reviewer’s comments here and partly agree. There is no doubt that aridification occurred multiple times with more humid periods in between during the last 33 million years. The main question is: did that lead to a single vicariant event (one event had a very strong impact) within clades, or multiple ones (several strong impacts)? Eg: an aridification event occurred, led to speciation/vicariance and then these distinct clades came into secondary contact and diverged again at the next aridification event, or never reconnected, ending with a single event.
The Davis et al. 2002 shows the case of a single vicariant event within Acridocarpus, even though aridification/humid phases occurred before/afterwards. In contrast, the Loader et al. 2007 paper suggests 2 events. In our Couvreur et al. 2008 paper, we suggested at least three. However, we agree that evidence has been in favor of the multiple impacts hypothesis (which was explicitly written in our first version), but then again it might just be a question of scale (the larger the clade, the more we shall find evidence for multiple events).Also, there are not that many studies explicitly testing these hypotheses, so it is hard to make this a generalisation but our study moves us closer to this goal.
In response to your comment we have added to our hypothesis section as suggested and focused on the “how many” and if these are congruent with other studies and aridification events in general. We maintain the two major hypotheses (we believe it remains a valid and structured way to set the stage) but add, as suggested, that the multiple events hypothesis has been favored and thus we mainly seek to infer how many and when these events happen and if they are congruent with other studies.
We added edited and add a sentence (Line 84, underlined below) and removed the last sentence of the associated paragraph
“Most other molecular dating studies published to date also support the hypothesis of multiple fragmentations in plant and animal clades which suggests a general pattern [34,35]. By focusing on an AFR restricted plant clade of the Annonaceae family, we seek in this study to infer how many vicariant events occurred, if these are congruent with known aridification periods and if these events coincide with other independently estimated events in other ARF clades.”
Finally, we adopted the last aim of our intro to (Line 118): (3) How many vicariant events do we infer within this clade, and are these events congruent with known climatic and geological events?
Line 76: Rather than “Other molecular dating studies”, I would say “Most other molecular dating studies”.
Done
Line 86: please, change “markers based a tailored” to “markers from a tailored”, please.
Done
Lines 89–90: please, remove space between frame and work so that it reads as “framework”.
Done
Line 93: “provide an” reads weird, maybe change to “present an”.
Done
Lines 106–107: as previously mentioned, rather than one versus many I would ask how many.
Done
Material and Methods
Line 116: why so many taxa in the outgroup?
In Annonaceae, we need to sample outgroups all the way to the crown node of the family for fossil calibration to ensure that we have captured the full evolutionary history of the group. Thus in most Annonaceae dated studies, there are a lot of outgroups, which doesn’t pose a problem in general and the wider sampling can improve the accuracy of calibration via node dating.
Line 129: why HybPiper 1.2? Version 1.3.1 has been around for over a year now and has greater capabilities as well as takes care of a number of bugs. It really doesn’t take that long to rerun this pipeline.
As suggested we have rerun the whole analysis using HybPier 13.1. This led to very few differences.
Lines 131–132: Other than Gblocks (please change spelling to match this one), why not use other tools, such as TreeShrink (Mai & Mirarab 2018, BMC Genomics), to automatically identify and remove possible outliers?
We followed your suggestion, ran TreeShrink in our 75/75 trees and have added the relevant passages in the text (Line 157).
Lines 132–137: wording in this sentence is confusing. Did you or did you not use the approach here described to remove potential paralogs from downstream analyses? Please, rephrase to remove ambiguity, thank you.
Done
Lines 143–147: what thresholds did you use when applying SortaDate to your 75/75 data matrix? Also, why not separate introns from exons? That is, divide supercontigs into exons, on one hand, versus introns, on the other (since nucleotide substitution models are probably very different between these two) and rerun ModelTest-NG on them.
We didn't use any thresholds here, but instead took the 32 most clock-like genes. In addition, we separated exons from introns as suggested (see changes in the text (parts 2.2 and 2.3 of Material and Methods)
Line 157: why use a uniform prior? Uniform priors are quite biased as they give equal importance to all values for a given interval.
Pirie and Doyle (2012) provided upper and lower limits for the crown age of Annonaceae. As we had information on these limits, and no information on where in the interval the divergence most likely occurred, we used a uniform prior. In this case we give equal weight to all values, limiting them by our prior knowledge while avoiding the biases other types of priors may introduce.
Speaking of which, what interval are you exploring? What’s your mean? And your standard deviation?
Our phrasing might have been misleading or unclear. We placed a uniform prior (so no mean or SD needed) between 89 and 112 Ma for the crown node of Annonaceae. We have made this clearer by explicitly stating the ages of our uniform prior.
Line 175:
As there is no other reliable fossil for Annonaceae, and no evidence of Piptostigmateae fossils, we choose to constrain the Annonaceae crown node between those dates (89-112 Ma) and apply a uniform prior following [58].
A normal prior is less biased when imposing this type of constraint. Why not use a secondary calibration on your root node and other maximally supported nodes (away from the tips)? E.g. those corresponding to topological constraints (lines 157–159).
Secondary calibrations are indeed good solutions in the total absence of fossil data, which is not the case in Annonaceae. As indicated above, dating the Annonaceae has been ongoing for over a decade, and we have calibrated our tree using fossil evidence, thus it is not appropriate to us a normal prior.
Lines 159–161: if you do model selection for each of your supercontigs (please, divide exons from introns and rerun ModelTest-NG), why aren’t you doing any model selection here?
We run the last released Model Selection tool implemented in BEAST2, which is Nested Sampling (NS). Changes have been made in the text (line 179), and results are presented in table S7.
You have used SortaDate to select genes likely evolving under a strict clock and then you run analysis under a relaxed molecular clock model (by the way, what distribution do you impose on this relaxed clock? Is it lognormal by any chance?). I’m confused. You should test different clock and tree models (other than under a pure birth model—that is, a Yule process—, one can also model speciation under a birth-death process).
Indeed, we selected genes that approach the strict clock, but that are not per se strict clock. We just try to minimize the variation across the genes we sample so we end up with simpler models to use when dating. Because we have so many genes we can afford to do this “shopping”. So no, the genes are not strict. However, we agree with the reviewer that is was not clear. We have added the Ln values of our strict versus relaxed clock models, which show that the relaxed model is favored.
If your time-calibrated phylogeny does not contemplate extinction (Yule process), I am not surprised you aren’t finding any later on. You can easily use path sampling (PS) and Stepping-stone Sampling (SS) to estimate marginal likelihoods (MLEs) and select the best supported model (which could be a strict clock model and a BD speciation process instead of what you have arbitrarily chosen; and if it isn’t arbitrary, please, properly justify it).
See above
Lines 163–166: I am surprised you consider ESS > 100 adequate. By default, Tracer has the threshold set at 200. Though, as long as mixing is acceptable, this shouldn’t be an issue. However, you do not explicitly state whether mixing is any good. Is mixing any good?
All ESS values were in fact bigger than 200 for the tree likelihood of each alignment (except one, with 65 ESS). The values under 200 ESS were for some parameters of the substitution models like the proportion of Invariants.
Line 167 (and line 229): please, change “Maximum Clade Credibility Tree (MCCT)” to “Maximum Clade Credibility (MCC) tree”, which is the convention in the existing literature (see any paper by the developers of BEAST), and MCCT to MCC tree in all subsequent mentions of this acronym (e.g., lines 174 and 224), thank you.
Done
Line 168: please, change “density (HPD) and” to “density (HPD) interval and”.
Done
Lines 172–174: this is why I do not understand why you have such a large outgroup. Please, explain.
As indicated above, we prefer to date the Annonaceae phylogeny using direct fossil evidence. This does not impact the analyses.
Also, why not try a clade dependent diversification approach such as MEDUSA (available in R package Geiger) that doesn’t require complete sampling?
BAMM doesn’t require a complete sampling either but has the advantage of identifying shifts continuously along branches, rather than at nodes as MEDUSA does. We prefer to leave the BAMM analysis and not add a new one.
Line 181: could you have some sort of supplementary table summarizing parameter exploration, please?
We test several priors between 0.1 and 1 for speciation and extinction, resulting in negligible changes to the results (0.0001 difference). We added a sentence in the text. (line 201)
Lines 182–187: this is why I do not understand why no model testing is done for divergence time estimation. I feel your tests are circular. When inferring divergence times, you impose a Yule process and then you get pure birth with no extinction as your most optimal result. Also, why exponential rates? Are there no other options?
We addressed the issue of selecting clock-like genes above. These are not strict clock genes, but the closest to strict clock in our dataset. We agree that this appears circular, but in fact studies have shown that the selection of tree or clock priors don’t affect the inference of diversification rates (see Sarver et al 2019). This of course, as suggested by the reviewer, doesn’t mean we can just choose these randomly, because they do impact age estimate (we have thus added formal tests). However, focusing on a set of clock “like” genes does not (should not) impact diversification analyses and thus our conclusions.
Sarver, B.A.J., Pennell, M.W., Brown, J.W., Keeble, S., Hardwick, K.M., Sullivan, J., Harmon, L.J., 2019. The choice of tree prior and molecular clock does not substantially affect phylogenetic inferences of diversification rates. PeerJ 7, e6334. https://doi.org/10.7717/peerj.6334
Line 194: You are clearly familiar with Stepping-stone Sampling (SS) to estimate MLEs for model selection (see comment to lines 159–161), why not use it to select the best fit model for divergence time estimation?
See above
Lines 195 and 205: I find 100 a suboptimal ESS threshold. Could you comment on the mixing?
We also verified mixing with a visualization of trace and density of the MCMC. (we add that in the text)
Line 205: how were Bayes Factors calculated?
We use the computeBayesFactors command in BAMMtools. We added in the text (line 227), see also new table S6.
Lines 207–208: why not use the approach proposed by Culshaw et al. 2019 (doi: 10.3929/ethz-b-000348893), which relies on Bayesian birth-death skyline (BDSKY) models, to detect mass extinction events (instead of TESS)? This BDSKY approach performs better than CoMET for smaller phylogenies and it is available in BEAST2.
In the paper cited by the reviewer it is said (in the abstract and main text):
"while CoMET performed better in detecting and locating MEEs for smaller phylogenies", and also "CoMET performed better under moderate/low-intensity mass extinction scenarios (ρ = 0.5, 0.9) and with smaller phylogenies (N = 100, 200)".
Therefore our use of CoMET is appropriate for our relatively small phylogeny. We have now mentioned this in the methods section line 233:
“This method has also been suggested to perform better with small phylogenies, as in our case, when compared to other methods such as BDSKY (Culshaw et al. 2019).
RESULTS
Line 382: please, change “Highest Posterior Density (HPD)” for “HPD interval”. You already explain this abbreviation elsewhere. No need to repeat it here.
Done
Lines 240–241 and 248–250: I do not see the point of this table in the main text. Most of that information is already available in your chronogram and the text. Please, move it to supplementary materials or completely remove it.
We have moved this to sup mat as suggested.
Lines 252–255: how is this not circular? Yule chronogram, yule diversification rate.
See our response above.
Line 265, 281, and 391: please, change “COMET” to “CoMET”, thank you.
Done
Lines 265–266: I understand CoMET struggles with small phylogenies, such as yours, and I urge you to try a BDSKY model in BEAST2 instead.
See our response above.
DISCUSSION
Lines 293–294: you need to separate exons from flanking partial introns (now together in supercontigs).
Done (see changes in the text)
Lines 295–297: your “gene shopping approach” (SortaDate) picks “genes” most congruent with the species tree, so this isn’t surprising in the least. I do not understand why it is highlighted here. Maybe change wording to reflect this expectation, given the chosen approach for divergence time estimation.
We did not follow the whole SortGenes steps, and in particular we did not select genes based on phylogenetic pattern, exactly so we could infer non biased relationships. We see this wasn’t clear and have added this sentence Line 164:
“However, we did not apply the phylogenetic relationship filter, i.e. we did not select genes that produced a particular phylogenetic relationship.”
Lines 305–306: please, note that your choice of priors could also lead to an increase in precision.
We chose a uniform prior, which as explained above places equal weight across the range of dates used. Thus, the increase in precision appears to be inked to our phylogenomic approach, rather than the choice of the prior as implied.
Line 310: this date coincides with the stabilization of the East Antarctic Ice Sheet (EAIS), see Flower & Kennett 1994.
We have added this comment and the reference as suggested.
Line 311: change “Ma) and followed shortly after the Middle Miocene Climatic Optimum (MMCO)” to “Ma), which followed the Mid Miocene Climatic Optimum (MMCO)”.
Done
Lines 313–314: in addition to the uplift of the African Plateau and the final closure of the Tethys sea, distant geological events, such as the previously mentioned stabilization of the EAIS, also played a role in the aridification of the Congo Basin (and no doubt in the global cooling mentioned in line 312).
Indeed, we have added this to the discussion.
Line 327:
In addition to these intra African events, other factors such as the stabilization of East Antarctic Ice Sheet could also have led to increased aridity in Africa Flower & Kennett 1994).
Lines 315–318: in addition to Dimitrov et al. 2012, Pokorny et al. 2015, Tosso et al. 2018, and Makga et al. 2020, see studies on legume subfamily Detarioideae by de la Estrella et al. 2017 (doi: 10.1111/nph.14523) and Choo et al. 2020 (doi: 10.1016/j.ympev.2020.106752), for further evidence.
We have added these citations as suggested.
Line 336: I would rephrase “Greenwayodendron were around this period” so that it reads “Greenwayodendron match this period” instead.
Done
Line 340: change “speciation date in” to “speciation timing in”, please.
Done
Line 345: In addition to Morley 2000, Linder 2017, and Jacobs et al. 2010, I would also cite Sepulchre et al. 2006 and Senut et al. 2009 in here.
Done
Line 354: remove space after “time” and before comma, please.
Done
Lines 365–367: which is why I am recommending the approach proposed by Culshaw et al. 2019.
See our comments on this above. From our readings we used the correct approach for small phylogenies.
Line 368: a MEDUSA-like approach could maybe pick up this clade-specific shift.
Yes, indeed, but so does BAMM.
Line 382–383: See Piñeiro et al. 2019 (doi: 10.1101/811463) for a recent study on refugia at the population level (as well as your own Helmstetter et al. 2020, doi: 10.1101/807727).
As discussed before, these studies refer to intraspecific diversification, and out of the scope of this macroevolutionary study.
Line 386–388: regarding your RPANDA analyses, maybe speciation is neither constant nor exponential, but you are not contemplating any intermediates between these two extremes (and they are quite the extremes, if I may say). Thus, I would not go as far as to say speciation is exponential.
This is a good point. We added this caveat to our discussion as suggested.
Line 400:
“However, because we did not test any intermediate models between exponential and constant, the true speciation rate probably lines somewhere in between these two extremes.”
Lines 395–397: see also the aforementioned de la Estrella et al. 2017.
We have cited this reference as suggested.
Lines 399–403: in Pokorny et al 2015 one can see that speciation and extinction are clade dependent (see figure 5 in that paper). So, neither high extinction nor high speciation but probably a number of different scenarios depending on the clade at hand and its tolerance to aridification-tropicalization shifts. The picture is a lot more complex than you are painting it to be.
We agree with this comment. However, the point here was to focus on diversification rates specifically, because that is one of the main hypotheses put forward to explain differences in diversity between major TRF regions. Of course we agree that processes are clade specific, but to explain a general pattern one should try and find common processes. Moreover, we do not suggest that speciation or extinctions rates are similar across clades, but that in general the “odd man out” pattern could be the result of simply lower speciation rates over all (versus higher ones outside of Africa).
Lines 405–406: but see de la Estrella et al. 2017.
Done
Lines 419: please, change “low extinction rate within African clades” to “low extinction rate within some African clades”.
Done
Line 415: see Choo et al. 2020 for an example in legume genus Daniellia.
Cited.
Lines 420–422: I repeat, you are not testing any intermediate options between constant and exponential speciation.
We have added a caveat in the discussion as suggested (see above).
Reviewer 2 Report
Main Comment
This study studied 23 species in Piptostigmateae out of 31 species and 7 species in Annickieae out of 8 species, and analyzed the divergence time using 32 exon sequences applying a Bayesian approach. For Piptostigmateae, model simulations were also conducted to assess the posted hypotheses of the recent diversification. Molecular data of this study seems to be based on the previous work Couvreur et al. (2019), but the conducted statistical analysis/simulation are much sophisticated to reveal the consequence of recent events of speciation. The manuscript is overall well written and will provide relevant information for the readers related to the biodiversity and evolutionary phylogeny. It is interesting that the molecular data supported the trend which the speciation rate increased with time, without extinction event. Authors concluded that “speciation mainly took place during the Pliocene and Pleistocene”(l.28-29) and “…is harder to link to a rain forest vicariance event”(l.413-414). If so, there could be the alternative explanation of the recent divergence; I guess variation/divergence of the morphology (flower is mainly assumed) or some other traits is a candidate viewpoint to be argued. It is because authors showed Fig.2 B-K which inferring the morphological divergence. I expect the authors will discuss more about the required result.
Furthermore, although author stated lower diversity in African rain forests, compared to those of Neotropics and South East Asia in Introduction Part, there was insufficient statement about the reason of the difference of the diversity in Discussion Part. I also expect the revision including what authors think about this difference.
Comment
l.26-27; “pure-birth (no extinction) exponential speciation rate” is just the name of the option of RPANDRA. For the readers’ easy comprehension, a brief explanation of the model should be stated here, instead of the current statement.
l.30; I wonder the former three keywords are appropriate for the critical words standing for this manuscript.
l.72-73; Please show briefly about the approach of the study of the referred article. They applied molecular analysis, I guess, is it right?
l.77-78; Please add the required or efficient approach to test the hypothesis.
l.100 “between these genera and species”; A comparison objective is ambiguous (genus level or species level?).
l.135-137; This sentence is hard to understand for me. Especially, what is the meaning of the “clade” here?
l.139-140; This step seems to be important, but the statement of this step is not consistent throughout the manuscript (l.140, l.142, l.218).
l.143-146; The applied “gene shopping” approach should be explained more. The explanations of the approach and the significance/validity are unclear.
l.175; Refer R.
l.219-221 or Table S3; There are no information about the range of the length of 32 contigs.
l.258-262; I cannot confirm the validity of the BAMM result, because there are no comparable values like Table 2 and Table S4. However, in the M&M, it is stated that “BF were then calculated to find the best fitting-model” (l.205-206).
l.397-408; In this part, authors referred to the previous study regarding to the “odd man out” pattern. As I mentioned above, I expect further discussion with adding the present result.
Check the consistency of the use of abbreviation word, because there are some spell-miss.
Fig. 2; I suggest that authors need to revise and re-organize this figure. Panel A is too small to check and evaluate the result. Please use bold line for the variance of each node. Add an explanation for the red fond. Add “Pleistocene” at the proper box. The panel A and panels B-K should be arrayed vertically.
Table S2; Please modify the order of samples according to the Table S1 (for easy understandings). What are the meanings of values in “number”, “INDEX”, “10x coverage”, and “maen depth” in Table. For later two, why authors used “,” (instead of “.”)?
Author Response
Point by point reply to the Review Report (in bold)
This study studied 23 species in Piptostigmateae out of 31 species and 7 species in Annickieae out of 8 species, and analyzed the divergence time using 32 exon sequences applying a Bayesian approach. For Piptostigmateae, model simulations were also conducted to assess the posted hypotheses of the recent diversification. Molecular data of this study seems to be based on the previous work Couvreur et al. (2019), but the conducted statistical analysis/simulation are much sophisticated to reveal the consequence of recent events of speciation. The manuscript is overall well written and will provide relevant information for the readers related to the biodiversity and evolutionary phylogeny. It is interesting that the molecular data supported the trend which the speciation rate increased with time, without extinction event. Authors concluded that “speciation mainly took place during the Pliocene and Pleistocene”(l.28-29) and “…is harder to link to a rain forest vicariance event”(l.413-414). If so, there could be the alternative explanation of the recent divergence; I guess variation/divergence of the morphology (flower is mainly assumed) or some other traits is a candidate viewpoint to be argued. It is because authors showed Fig.2 B-K which inferring the morphological divergence. I expect the authors will discuss more about the required result. Furthermore, although author stated lower diversity in African rain forests, compared to those of Neotropics and South East Asia in Introduction Part, there was insufficient statement about the reason of the difference of the diversity in Discussion Part. I also expect the revision including what authors think about this difference.
Indeed, we do not discuss the odd man out in detail here. We didn’t want to talk too much about this because to really test this pattern a pan tropical sampling is needed. However, we feel that our result provides some limited addition to the discussion. Thus we mention it briefly here as not to over emphasize our results given the intra Africa scope of our study.
Comment
l.26-27; “pure-birth (no extinction) exponential speciation rate” is just the name of the option of RPANDRA. For the readers’ easy comprehension, a brief explanation of the model should be stated here, instead of the current statement.
Changed as suggested: “one with exponential speciation rate and no extinction”.
l.30; I wonder the former three keywords are appropriate for the critical words standing for this manuscript.
We added the word hypothesis.
l.72-73; Please show briefly about the approach of the study of the referred article. They applied molecular analysis, I guess, is it right?
Yes this is correct, we have added that statement to the start of the sentence.
l.77-78; Please add the required or efficient approach to test the hypothesis.
We have added this sentence as suggested:
Line 91
“In order to test these hypotheses and infer the age of vicariances, we can use dated molecular phylogenies of clades spanning east and west/central Africa, with endemic species in both regions (cite Loader et al. 2007 and Couvreur et al. 2008).”
l.100 “between these genera and species”; A comparison objective is ambiguous (genus level or species level?).
We have removed the reference to genus, in the end they are all species, except that in some case genera are monospecific.
l.135-137; This sentence is hard to understand for me. Especially, what is the meaning of the “clade” here?
We have reworded this sentence, see line 146.
l.139-140; This step seems to be important, but the statement of this step is not consistent throughout the manuscript (l.140, l.142, l.218).
We have made this step consistent as suggested.
l.143-146; The applied “gene shopping” approach should be explained more. The explanations of the approach and the significance/validity are unclear.
We have added additional explanation of the gene shopping approach (Line 159)
l.175; Refer R.
Done
l.219-221 or Table S3; There are no information about the range of the length of 32 contigs.
We added the length of the exons/intron in the table.
l.258-262; I cannot confirm the validity of the BAMM result, because there are no comparable values like Table 2 and Table S4. However, in the M&M, it is stated that “BF were then calculated to find the best fitting-model” (l.205-206).
We add one more table in supplementary materials. (table S6)
l.397-408; In this part, authors referred to the previous study regarding to the “odd man out” pattern. As I mentioned above, I expect further discussion with adding the present result.
See above.
Check the consistency of the use of abbreviation word, because there are some spell-miss.
Done
Fig. 2; I suggest that authors need to revise and re-organize this figure. Panel A is too small to check and evaluate the result. Please use bold line for the variance of each node. Add an explanation for the red fond. Add “Pleistocene” at the proper box. The panel A and panels B-K should be arrayed vertically.
We have followed the reviewer’s suggestion. We made a figure only of the phylogeny and placed the photos in a separate figure together with a new figure 1.
Table S2; Please modify the order of samples according to the Table S1 (for easy understandings). What are the meanings of values in “number”, “INDEX”, “10x coverage”, and “maen depth” in Table. For later two, why authors used “,” (instead of “.”)?
We remove the “INDEX” and “TAG” column as there are not relevant to this study. We explain the 10x coverage ane mean depth in the title of the table.
Reviewer 3 Report
see comments in the PDF file (attached). My main concern is that the map of the modern distribution of African wet forests presented in the Introduction seems not to be relevant to the Results and Discussion text. I have minor other concerns highlighted with comment (if without comment then perhaps a misspelling or grammatical issues is noted).

Author Response
Point by point reply to the Review Report (in bold)
We made all typo/edits suggested by the reviewer in the PDF. We respond to the comments below:
The word "major" refers to what? Species-rich? Ecologically dominant? Are the Annonaceae and any of its constituent subclades sufficiently species rich so as to address diversification issues?
The word major was supposed to mean that Annonaceae are a dominant plant family of lowland rain forests, generally in the top ten diverse families. We have change major to
Line 16: “ecologically dominant and diverse tropical rain forest plant family”
Do singular historical events explain modern patterns of biodiversity rather than ongoing ecological processes?
We agree with the reviewer’s point of view, both historical and ecological factors can be at play. Here we explore in particular the historical side.
What does the green to the NE represent. Is the arid corridor connected at the N end?
We have rearranged figures 1 and 2. The initial forest cover map presented all forested regions across tropical Africa, which was not appropriate given our focus. We have replaced that map with a map of tropical rain forests (taken from Couvreur et al. 2008) so this comment is no longer applicable.
Several genera endemic to each of these two regions?
We have rephrased this sentence:
Line 61
“In addition, each region is home to several endemic genera.”
The Results and Discussion do not clearly address the relevance of this modern distribution of African wet forests. Perhaps the modern distribution of these forests have no relevance because of historical influences?
See our response above. We changed the figure and this comment is no longer applicable.
Line 94: Wording is unclear here and the highlighted sentence below.
We deleted that sentence.
We changed the last sentence of this paragraph to:
Line 111:
“Estimating speciation events within this tribe between east and west/central African endemics and correlating these with known abiotic events responsible for rain forest fragmentation (i.e. aridification events) can improve our understanding of ARF evolution.”
Line 111: Sampling is limited. Why not multiple conspecific samples and closer to 31 species sampled?
Multiple samples per species increases computation time and cannot be included when using methods such as BAMM, CoMET.. It was shown in Couvreur et al. 2019 that the samples we used were good species representatives. We sequenced all available species, to get as close to 31 as possible. Material is not available for the remaining species. . We have rewritten his sentence to make it clearer what we did:
Line 122:
“We sequenced one individual per species for the 23 available species, out of the 31 species known in Piptostigmateae (74%), covering the six genera of the tribe, and seven species of eight from the genus Annickia (Annickieae).”
Line 115: 3 + 27 = 30 species endemic to specified regions. Does this include ingroup and outgroup species?
Apologies for the misunderstanding. It is as you said 27+3=30 endemics species, plus 23 species added as outgroup, which makes 53 species in total. We change the sentence in the text to be clearer about it (line 125).
Line 128: 23 ingroup plus 23 outgroup species = 46 total. Why 53 here?
See previous comment
157: Is this a uniform distribution between 89-112 Ma? Shouldn't the Annonaceae crown and stem clade each be constrained using something like an exponential density calibration prior with an offset at 89 and 112 Ma, respectively?
riors on calibrations are quite subjective. In the case of this calibration, we have no prior information to prefer a uniform over exponential (or normal as suggested by Rev 1). Here we follow the main authority of the subject, and the hypothesis used for Annonaceae in past years: Pirie & Doyle 2012. Uniform priors place equal weight across the age limits , and are thus the least subjective of prior distributions.
Line 197: Are any of the clades in question with regard to rate shifts that species rich?
This is a fair and interesting point. However, rate shifts are not only related to species richness (eg more than 30 species). They are a function of time, species richness and also relationships to other clades (species rich or poor clades?). As such any clade could potentially be detected as a rate shift thus the necessity to test this.
Line 238: Ages are reported without any context. What is the controversy? Are these ages needed for the diversification analysis?
This passage is in the results section, so we do not give context until the discussion. We discuss the context briefly in the discussion. Indeed, these ages are not useful for the objectives of our study, but this is the first time a phylogenomic nuclear dating analysis has been done within Annonaceae (versus plastid dating). It is thus interesting to report them here. We have added this mention to the second paragraph of the discussion.
Line 308:
“Our dating analysis is the first time numerous nuclear markers were used to date clades within Annonaceae. Past studies have mainly relied on a few plastid markers (e.g. Richardson et al. 2004; Guo et al. 2020) or full plastomes (Migliore et al 2019; Lopes et al. 2018)”.
366: Why were not more species sampled?
This would have been great, and was our objective but no material was available for some species or the DNA extraction didn’t pass the quality step.
Line 408. Thus, these results are not robust and not worth addressing in the first place.
We understand the viewpoint of the reviewer to a certain extent. However, we do think these results are worth reporting, given the caution we provide. Our results are based on the data at hand and using the best available methods. It is within that context that we interpret them.